# Immunohistochemistry Reveals TRPC Channels in the Human Hearing Organ—A Novel CT-Guided Approach to the Cochlea

**DOI:** 10.3390/ijms24119290

**Published:** 2023-05-26

**Authors:** Colya N. Englisch, Jakob Steinhäuser, Silke Wemmert, Martin Jung, Joshua Gawlitza, Gentiana Wenzel, Bernhard Schick, Thomas Tschernig

**Affiliations:** 1Institute of Anatomy and Cell Biology, Saarland University, 66421 Homburg/Saar, Germany; 2Department of Otorhinolaryngology, Head and Neck Surgery, Saarland University, 66421 Homburg/Saar, Germany; 3Institute of Medical Biochemistry and Molecular Biology, Saarland University, 66421 Homburg/Saar, Germany; 4Institute of Radiology, Technical University of Munich, 80333 Munich, Germany

**Keywords:** TRPC6, TRPC5, TRPC3, human cochlea, organ of Corti, temporal bone, body donors, immunohistochemistry, computed tomography

## Abstract

TRPC channels are critical players in cochlear hair cells and sensory neurons, as demonstrated in animal experiments. However, evidence for TRPC expression in the human cochlea is still lacking. This reflects the logistic and practical difficulties in obtaining human cochleae. The purpose of this study was to detect TRPC6, TRPC5 and TRPC3 in the human cochlea. Temporal bone pairs were excised from ten body donors, and the inner ear was first assessed based on computed tomography scans. Decalcification was then performed using 20% EDTA solutions. Immunohistochemistry with knockout-tested antibodies followed. The organ of Corti, the stria vascularis, the spiral lamina, spiral ganglion neurons and cochlear nerves were specifically stained. This unique report of TRPC channels in the human cochlea supports the hypothesis of the potentially critical role of TRPC channels in human cochlear health and disease which has been suggested in previous rodent experiments.

## 1. Introduction

Hearing loss is a common progressive neurosensory deficit affecting all life periods and especially the elderly [1]. It is a complex and often multifactorial process including injuries of the stria vascularis, of cochlear hair cells and of spiral ganglion neurons (SGNs) [2]. From a physiological point of view, it is however still not entirely understood which ion channels are involved in the mechano-electrical transduction of cochlear hair cells. Many ion channels have been proposed as candidates [3]. In any case, Ca^2+^ ions and respective molecular structures are of critical relevance [3]. Focusing on the Transient Receptor Potential Canonical family, TRPC1 and TRPC6 were found to be involved in mechanosensory complexes [4,5]. A double rodent knockout of TRPC3/TRPC6 was investigated and demonstrated hearing impairment and other functional phenotypes [6].

Transient Receptor Potential (TRP) channels belong to the family of non-selective cation channels [7]. The TRPC1-7 channels shape one of several described TRP subfamilies. They represent primary non-voltage-activated Ca^2+^ influx pathway in cells. For instance, their participation in receptor- and store-operated Ca^2+^ entry has been demonstrated [8]. TRPC channels appear as homo- or heterotetramers [9]. Each of the four monomeric subunits is composed of six transmembrane segments. The cytosolic COOH and NH_2_ termini feature different domains, including coiled-coils, ankyrin-like repeats, a TRP-box and a calmodulin and IP_3_-receptor binding (CIRB) site, ultimately enabling broad interactions with other molecular players [8,10]. TRPC channels are subject to a diversity of regulating mechanisms that are described in detail elsewhere [11,12]. They are widely expressed [13] and often critically involved in pathologies such as, for example, ischemia/reperfusion injuries or tumorigenesis [11,14,15]. However, the role of TRPC channels in the physiology and pathophysiology of the auditory system has not yet been fully clarified.

The cochlea, where sound coding takes place through mechano-electrical transduction, is embedded in the petrous part of the temporal bone. The organ is approximatively 35 mm long and is twisted 2.5 times around the modiolus—the bony axis of the cochlea [16] (Figure 1). The spiral lamina of the cochlea is a bony ridge emanating from the modiolus in a centrifugal manner. The Reissner’s and basilar membranes arise from this bony ridge to divide the cochlea into three different ducts: the scala vestibuli, the scala media and the scala tympani (Figure 2). The stria vascularis is localized in the lateral wall of the scala media and is a host to multiple molecular players that are responsible for the production of the cytosol-like endolymph in terms of ionic composition [17].

Hearing is dependent upon mechanosensitive hair cells (HCs) that are localized to the organ of Corti. Vibrations of the basilar membrane induce deflections of HC stereocilia-bundles, ultimately activating not yet fully identified mechano-electrical transducer channels [3]. Cation influx results in both inner (IHCs) and outer hair cells (OHCs) and generates a depolarizing receptor potential. While IHCs stimulate afferent fibers of SGNs through the Ca^2+^-dependent activation of the ribbon glutamate exocytosis machinery [18], OHCs can indirectly increase IHC stereocilia deflections and finally boost the neural signal. Detailed summaries of the mechanisms necessary for normal auditory function are presented elsewhere [16,19].

The aim of this study was to prove the existence of TRPC6, TRPC5 and TRPC3 in the human cochlea and to describe their respective expression profiles. For this, blocks containing cochleae were removed with the aid of computed tomography (CT) from the temporal bones of ten embalmed body donors. Samples were then exposed to 20% EDTA (ethylenediaminetetraacetic acid) decalcification solutions for several months. After embedding, immunohistochemistry was performed on the cochlea sections using anti-TRPC6, anti-TRPC5 and anti-TRPC3 knockout-validated antibodies. Myosin VIIa-staining was also performed to highlight the HCs.

**Figure 1 ijms-24-09290-f001:**
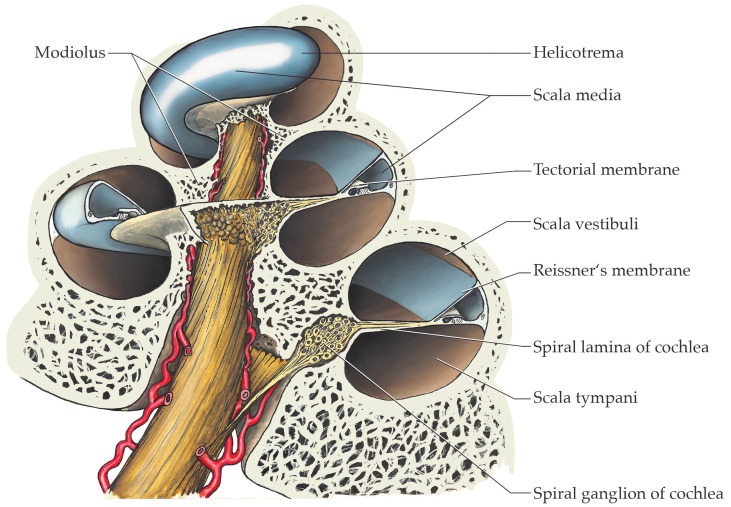
Overview of the anatomy of the cochlea in the petrous part of the temporal bone. Modified from Functional Histology, E. Lütjen-Drecoll, Stuttgart: Thieme; 2021 [20].

**Figure 2 ijms-24-09290-f002:**
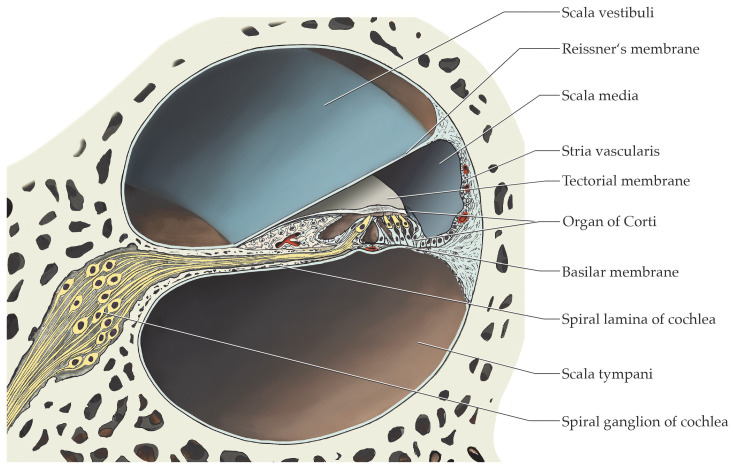
Cross section of the duct system of the cochlea. The scala media contains the organ of Corti with yellow-stained inner and outer hair cells, accompanied by different blue-stained supporting cells. Modified from Functional Histology, E. Lütjen-Drecoll, Stuttgart: Thieme; 2021 [20].

## 2. Results and Discussion

### 2.1. TRPC6, TRPC5 and TRPC3 Are Revealed in Sections of the Human Cochlea

The structures of interest included the spiral lamina of cochlea, Reissner´s membrane, the basilar membrane with the organ of Corti and the tectorial membrane, cochlear nerves and spiral ganglions, as well as the stria vascularis. Despite different factors such as prolonged postmortal intervals or increased chemical stress during the decalcification procedure, several anatomical structures remained well preserved. From a total of 15 specimens, the organs of Corti, striae vasculares, spiral ganglions, cochlear nerves, spiral laminae of cochlea, and basilar, tectorial and Reissner´s membranes were identified in 7, 13, 4, 13, 7, 7, 9 and 8 cases, respectively. Our focus was on the detection of TRPC6 in these structures. A pink/red chromogen indicated TRPC6 expression, and staining was similar in all specimens. In addition, we exemplarily investigated TRPC5 and TRPC3 in three samples. Figure 3A presents the main areas of interest, with the scalae vestibuli, media and tympani, and demonstrates TRPC6 immunoreactivity in different localizations. The basilar membrane itself was not stained, even though the organ of Corti featured an immunoreaction against TRPC6 antibody (Figure 3B). The TRPC5 and TRPC3 staining was similar to that of TRPC6. Anti-myosin VIIa staining—used for hair cell demarcation [21]—was detected in isolated cells of the organ of Corti (Figure 3C). Due to their direct positioning on the basilar membrane, these can either represent dislocated HCs during a long conservation and tissue processing period or simply an unspecific stain precipitation in some supporting cells. Therefore, it seems that TRPC6, TRPC5 and TRPC3 are localized at least in supporting cells of the organ of Corti and possibly in HCs as well. Reissner´s and the tectorial membrane did not present clear immunoreactivity against TRPC6, 5 and 3 antibodies (Figure 3A,D). In contrast, the spiral lamina of the cochlea was clearly immunoreactive (Figure 3D). The stria vascularis was similarly positively stained (Figure 3A). Certain SGNs (Figure 4A,B) featured a stronger staining compared to cochlear nerve fibers (Figure 4C). Interestingly, not all SGNs were immunoreactive and some of them were only partly so (Figure 4A,B). Negative controls obtained by omission of the primary antibody exhibited no red staining at all in contrast to TRPC chromogen detection (Figure 4D). The TRPC6/5/3 antibody specificity was validated by weaker staining observed after blocking-peptide incubation.

### 2.2. The Cochlear Distribution Profile of TRPC Channels Is Similar in Rodents and Humans

Here we investigated the expression of the Transient Receptor Potential Canonical channels 6, 5 and 3 in the human cochlea. This covered the spiral lamina of cochlea, Reissner´s membrane, the basilar membrane with the organ of Corti and the tectorial membrane, the cochlear nerve, the spiral ganglion, and the stria vascularis. TRPC expression in different animals has been investigated in several studies in the past. For instance, TRPC3 mRNA was detected in rat and guinea pig organs of Corti using reverse transcription polymerase chain reaction (RT-PCR). Interestingly, in this study, the supporting and pillar cells were not labeled by TRPC3 immunofluorescence. However, IHC and OHC were comparably immunoreactive for TRPC3 [22]. In 2009, Takumida and Anniko investigated the expression of TRPC channels in the mouse inner ear using immunohistochemistry [23]. The immunofluorescence of TRPC6, TRPC5 and TRPC3 was accentuated in the stria vascularis, similarly to that in our human samples. In contrast to Raybould et al. [22], the IHCs and OHCs—but also supporting cells of the organ of Corti—featured immunoreactivity to TRPC6, TRPC5 and TRPC3. The murine spiral ganglion cell perikarya ubiquitously expressed TRPC6, TRPC5 and TRPC3. However, TRPC3 accumulated primarily in the cytoplasm, whereas TRPC6 featured an intense nuclear staining. TRPC5 was diffusely distributed. In contrast, it seems that certain human SGNs are only partly immunoreactive for TRPC6, TRPC5 or TRPC3 (Figure 4A,B). The cochlear nerve fibers were intensively immunoreactive to TRPC3, whereas only a slight immunofluorescence to TRPC5 and TRPC6 was detected, a phenomenon that was not validated by our investigation in human tissue. In the same period, Tadros et al. studied the distribution of TRPC3 in guinea pig cochlea [24]. In analogy to our sections, the SGNs featured the most intense immunoreactive signal. HC immunofluorescence was stronger in inner than in outer cells and more accentuated at the base of the cochlea compared to the apex. The expression pattern was qualitatively similar in murine cochleae [24]. TRPC6 and TRPC3 expression in cochlear HCs and SGNs was supported by further investigations [6]. In conclusion, it seems that animal investigations offer better research conditions, ultimately allowing for functional analyses and more detailed morphologic descriptions. However, the TRPC distribution profile in rodent cochleae seems to resemble human analogues, although small differences may exist—even between different rodent species.

### 2.3. TRPC Channels Are Critical Players in the Hearing Function in Rodents

As described above, TRPC channels are found in multiple localizations in the cochlea, including neuronal structures. Sensory neurons such as dorsal root ganglia are known to functionally express TRPC channels [25]. The role of these channels in sensation and nociception is becoming increasingly evident [26].

In terms of non-neuronal structures, we focused on HCs which are the cellular key-players in sound sensation [16]. Early investigations suggested that a TRPC3-like ion channel is involved in Ca^2+^ entry in the OHCs of guinea pigs and rats [22]. Ca^2+^ entry in the HCs was diminished upon TRPC3 knockout, as demonstrated by Ca^2+^ indicator microfluorimetry [27]. Indeed, cytosolic Ca^2+^ depletion is concomitant with decreased Ca^2+^/calmodulin levels and ultimately with reduced inhibitory complex binding to the CIRB domain of TRPC3, thus enhancing TRPC3-mediated Ca^2+^ influx [28]. Interestingly, OHC transduction in TRPC3^−/−^ mice is increased, as indicated by distortion product otoacoustic emission (DPOAE) measurements. In line with this, auditory brainstem response (ABR) thresholds are also lowered in TRPC3^−/−^ mice [27]. Moreover, cytosolic Ca^2+^ depletion can significantly modulate the K^+^ conductance—a key prerequisite in cochlear mechano-electrical transduction—of OHCs in a guinea pig model [27,29]. TRPC3 knockout-induced Ca^2+^ entry decrease is therefore associated with an increased OHC electromotility. Subsequently, boosted IHC transduction and auditory transmission can promote an hyperacoustic phenotype [27]. Altogether, TRPC3 is involved in the regulation of HC Ca^2+^ homeostasis and, ultimately, in the modulation of cochlear sensitivity. In other studies, single TRPC3 or TRPC6 knockout did not significantly affect the hearing function [6]. It is assumed that functional redundancy plays a critical role here. In agreement herewith, double TRPC3/TRPC6 knockout involved both auditory deficits and impaired ABR to high-frequency stimuli. Interestingly, mechano-electrical transduction of basal high-frequency sensing OHCs was reduced in this study [6]. Later, a quadruple TRPC1/TRPC3/TRPC5/TRPC6 knockout was introduced and tested [30]. Normal hearing function seems to require TRPC1 and TRPC5 as the quadruple knockout featured higher ABR thresholds compared to that of double TRPC3/TRPC6 deletion. However, the quadruple knockout had no impact on the mechano-electrical transduction (MET) currents from cultured OHCs [30]. Thus, TRPC deletion is likely to impair hearing function elsewhere than at the MET channel level. Moreover, the single-channel conductance of the HC MET channel is superior to that of TRPC channels [31]. All in all, hearing remains a complex function. When it comes to cellular sensation, different mechanisms, including MET, become relevant [3]. Several proteins shape the MET machinery. However, the exact players forming the MET channel are still unknown, even though some candidates, including LHFPL5 (Lipoma HMGIC fusion partner-like 5), TMIE (Transmembrane inner ear expressed protein) and TMC½ (Transmembrane channel-like proteins 1 and 2), have already been pre-selected. However, neither TRPC3 nor TRPC6 satisfy the criteria for the MET channel [3]. Their role in audition lies elsewhere.

### 2.4. TRPC Channels Are Potential Key Structures in Human Cochlear Health and Disease

Despite all the information collected, our study exhibits some limitations. Tissues gained from body donors, for instance, cannot entirely reflect the conditions in vivo. Parameters such as a prolonged postmortal interval or a long fixation procedure can damage the samples. In our case, the specimen is even further impaired by the decalcification procedure. Additionally, this study only describes the TRPC6/5/3 cochlear distribution in elderly subjects as this was the age group of the body donors. It might not necessarily reflect the circumstances in younger specimens. It remains unknown whether any of these body donors suffered from hearing deficits.

However, this report remains unique as it is—to the authors’ knowledge—the first time any TRPC channels have been described in the human cochlea. It is known that protein distribution profiles detected by immunochemistry or immunofluorescence vary between species [32,33]. Therefore, our study is of great interest as it finally confirms the localization of TRPC channels in different key structures of the cochlea. Based on previous functional reports gained from animal studies, we highlight here the potential relevance of TRPC6, TRPC5 and TRPC3 in human cochlear physiology and pathophysiology.

## 3. Materials and Methods

### 3.1. Body Donors: Approval, Sex, Age, Embalming and Preparation

Research based on body donations was approved by the Ethics Committee of Saarland Medical Association under application number 163/20 and conducted in accordance with the Declaration of Helsinki. All body donors had previously given their consent. Sex, age, and fixation method of the 10 body donors presented in this study are listed in Table 1. The mean age of the body donors was 84.2 years with a standard deviation of 8.2 years. The embalming methods, according to Basler or Weigner, are described elsewhere [34]. Circular incision of the skin was performed using anatomical landmarks, including the glabella ossis frontalis and linea nuchalis suprema ossis occipitalis. The musculi temporalis, procerus and occipitofrontalis were incised, and the skull was exposed. The skullcap was subsequently removed using an oscillating saw, hammer, and chisel. Preparation of the connective and nerve tissue followed thereafter to uncover the dura-covered cranial fossae. Anatomical reference points, including the internal acoustic pore with the vestibulocochlear nerve as well as the labyrinthine artery and vein, were explored to allow for a better orientation. The dura mater was subsequently removed in this area. Then, the petrous part of the temporal bone was excised in toto with CT guidance. For this, the distance from the lateral border of the cochlea to the medial border of the petrous part was first measured, and 5 mm were added thereto to create a sufficient safety distance (Figure 5A). The determined diameter of the cochlea and the thickness of the specimen were also given. In a second step, a fine-tuning of the big block was performed using a belt saw, and cochlear integrity was secured by CT scan (Figure 5B).

### 3.2. Imaging of Specimens

To allow for a precise preparation of the cochlea specimens, CT scans of the first skull were performed before preparation. Here, the size of the cochlea was measured in correlation to anatomical landmarks (e.g., squamous part of the temporal bone). The distances were considered similar among the skulls and excision procedure was therefore later performed without prior scanning. The respective cochlea specimens were scanned after dissection (Figure 5B). All scans were performed on a third-generation dual-source CT scanner (Somatom Force, Siemens Healthineers, Erlangen, Germany). The tube voltages were 80 kVp (Tube A) and 150 kVp (Tube B). In front of tube B was a dedicated tin filter for spectral shaping. Mean mAs was 120, rotation time was 0.25 s, pitch was 1.2 and detector collimation was 192 mm × 0.6 mm. All images were reconstructed using a relatively sharp kernel (Br69). Adjusted MRPs were reconstructed on the CT workstation. For objective measurements of the cochlea dimensions and precise positioning of the specimens after preparation, a 3D model (mesh volume) of the specimens was generated using a region-growing algorithm (ITK-Snap [35]; Figure 5C–F).

### 3.3. Decalcification Procedure—Challenges and Improvements

The petrous part of the temporal bone has the highest bone density in the human organism [36]. Decalcification is therefore challenging but necessary to achieve immunohistochemistry of the cochlea. Further difficulties, such as increased post-mortal intervals or the use of body donors for educational purposes, are detrimental to research and have a negative effect on the quality of the samples in terms of structural and molecular impairment.

Various decalcification methods are available [37,38]. However, the limitation of sample exposure to chemical stress is mandatory. Therefore, the mild but lengthy EDTA decalcification method was chosen. Each sample originally had a volume close to 8 cm^3^, to which 500 mL of 20% EDTA solution were added. The incubation time and frequency of replacements are listed below (Table 2). The solution was changed 34.9 times in mean to avoid chelator saturation and the mean duration of decalcification was 200.9 days or 28.7 weeks. The use of a microwave oven can dramatically accelerate the procedure [39]. Ghosh et al. were even able to carry out immunostaining within one month by regularly paring down extraneous bone [40]. In five cases of our study, only one side was eligible for histological investigation.

### 3.4. Immunohistochemical Tracing of TRPC6, TRPC5, TRPC3 and Myosin VIIa

The specimens were then embedded in paraffin, and 6 µm thin sections were prepared using frozen paraffin blocks and a microtome. After removing the paraffin and antigen rehydration, retrieval was performed using 1% citrate buffered solution for 15 min at 95 °C in a heating incubator, after which the samples were allowed to passively cool down for 30 min in the citrate. After washing the samples twice for 2 min in phosphate-buffered saline (PBS) each time, they were incubated with 5% BSA buffer (PBS) for 30 min. Then a knockout-validated antibody against TRPC channels (TRPC3/5/6) and myosin antigen structures was applied (Alomone Labs; cat. no. ACC 016, 017, 020, Jerusalem, Israel, Anti-Myosin VIIa/MYO7A antibody, Abcam ab230631, Cambridge, UK). The specificity and quality of the TRPC6 antibody was assessed using peptide-blocked control samples (Alomone Labs; cat. no. BLP-CC017). Same specificity control was also performed for TRPC5/3 but on renal tissue (Alomone Labs; cat. no. BLP-CC016/020). For TRPC6 antibody specificity, 40 µg of the control peptide was dissolved with 20 µL PBS and then incubated in tubes with 40 µL 1:10 diluted TRPC6 primary antibody overnight at 7 °C. The omission of the primary antibody was included in each staining run as a negative control. The protein concentration in both solutions was identical (0.01 mg/mL). Kidney samples that are known to express TRPC6 served as positive control [41]. The murine anti-cytokeratin AE1/AE3 antibody (MAB3412, Merck, Darmstadt, Germany) was also employed for positive control. Hematoxylin counterstaining was performed, and for detection of the antibody, a kit was applied (Labelled Polymer—Dako REAL™ Detection System, Alkaline Phosphatase/RED, Rabbit/Mouse, K5005, Agilent, Glostrup, Denmark) using a red chromogen.

### 3.5. Evaluation and Statistics

Slides were mounted and analyzed using light microscopy. Microphotographs were taken after slide digitalization using the Nano Zoomer S210 (C13239-01, Hamamatsu Photonics, Hamamatsu, Japan) and the NDP.view2 image viewing software (U12388-01, Hamamatsu Photonics, Hamamatsu, Japan). Evaluation was performed independently by two investigators. Statistical methods were not applied since the study was descriptive.

## Figures and Tables

**Figure 3 ijms-24-09290-f003:**
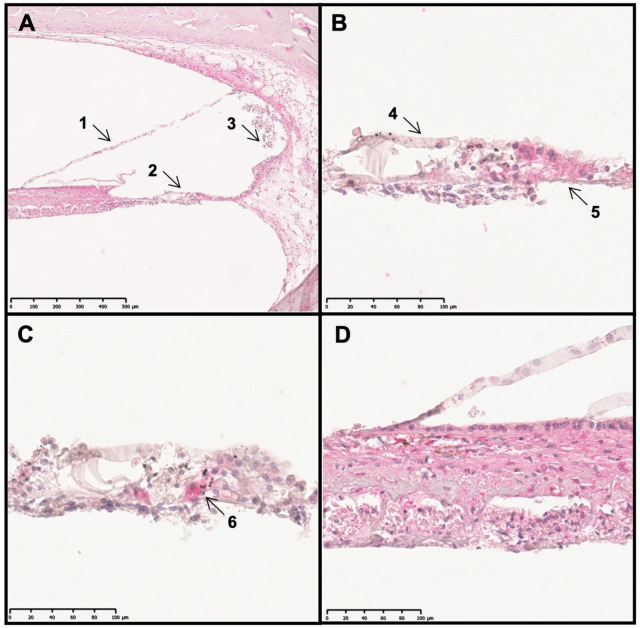
Distribution profile of different TRPC channels in the human cochlea. Pink/red chromogen indicates immunoreactivity. TRPC6 staining with a focus on the scala media (**A**). TRPC6 staining of the organ of Corti (**B**). Myosin VIIa staining within the organ of Corti (**C**). TRPC3 staining of the spiral lamina of the cochlea with Reissner’s membrane (**D**). 1 = Reissner’s membrane, 2 = organ of Corti, 3 = stria vascularis, 4 = tectorial membrane, 5 = basilar membrane, and 6 = potential hair cell.

**Figure 4 ijms-24-09290-f004:**
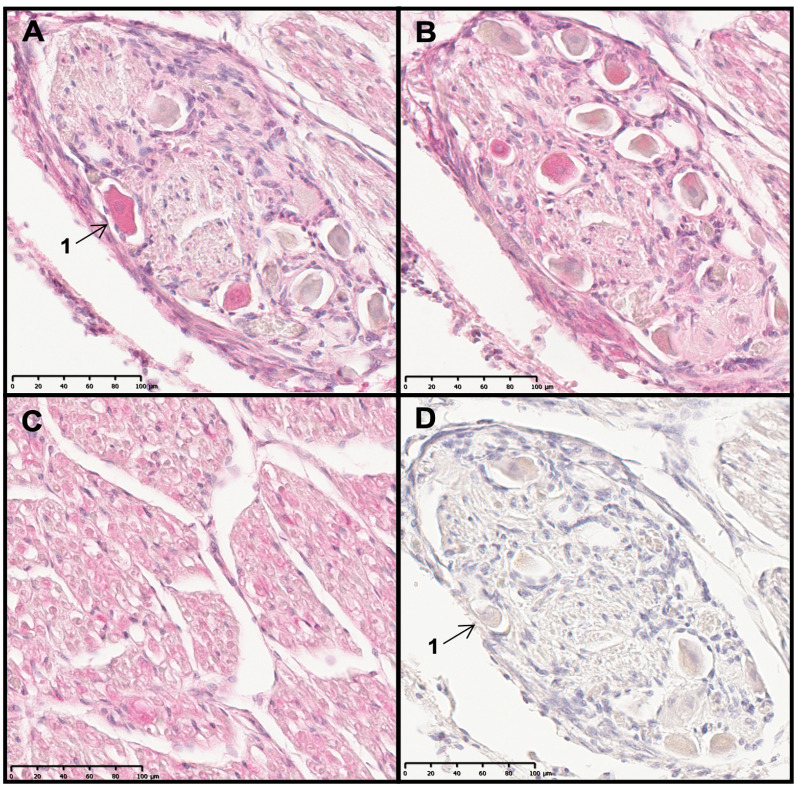
Distribution profile of different TRPC channels in the human cochlea. Pink/red chromogen indicates immunoreactivity. TRPC3 staining of spiral ganglion neurons (**A**). TRPC6 staining of spiral ganglion neurons (**B**). TRPC6 staining of cochlear nerve fibers (**C**). Negative control of the spiral ganglion neurons (**D**). 1 = spiral ganglion neuron.

**Figure 5 ijms-24-09290-f005:**
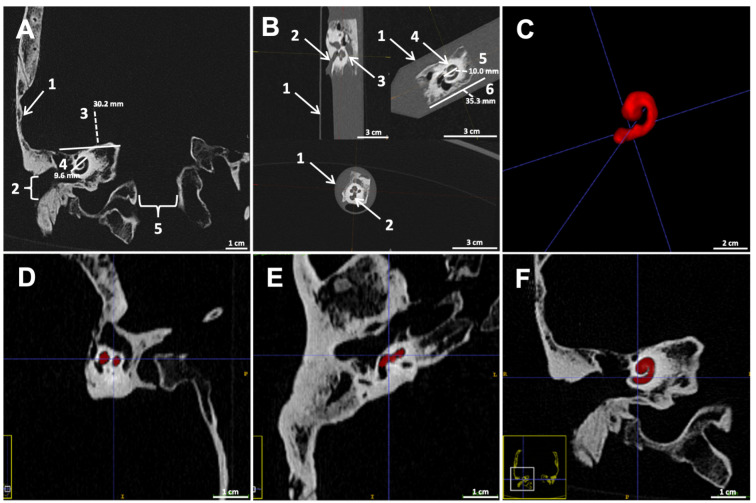
Computed tomography (CT) scan before preparation with exemplary measurements. 1: Squamous part of the temporal bone; 2: external acoustic pore; 3: distance between medial cochlea border and the tympanic membrane plane = 30.2 mm; 4: cochlear diameter = 9.6 mm; and 5: Foramen magnum (**A**). CT scan of the specimen after preparation inside a Falcon tube. 1: Falcon tube; 2: internal acoustic meatus; 3: semicircular canal; 4: cochlea; 5: cochlear diameter = 10.0 mm; and 6: block length = 35.3 mm (**B**). A 3D model of the cochlea, calculated from a 3D mesh volume by use of a region-growing algorithm after extraction and inside the CT images in axial, sagittal and coronal planes (**C**–**F**).

**Table 1 ijms-24-09290-t001:** Cadaver Characteristics.

Number	Sex	Age (Years)	Embalming
1	female	86	Weigner
2	female	84	Basler
3	female	71	Basler
4	male	70	Weigner
5	female	90	Weigner
6	female	93	Weigner
7	female	79	Basler
8	female	89	Basler
9	male	89	Basler
10	male	91	Basler

**Table 2 ijms-24-09290-t002:** Sample Decalcification Characteristics.

Number	Total Count Decalcification Solution Replacements	Total Time of Decalcification (Days)
1 left (l)	27×	93
1 right (r)	53×	270
2 l/r	51×	190
3 r	51×	193
4 r	40×	169
5 l	41×	172
6 l/r	30×	172
7 l	23×	211
8 l/r	23×	211
9 l	27×	253
10 l/r	27×	253

## Data Availability

Not applicable.

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
