# Peer review of "Immunohistochemistry Reveals TRPC Channels in the Human Hearing Organ—A Novel CT-Guided Approach to the Cochlea"

_ijms, 2023, doi:10.3390/ijms24119290_

Round 1

Reviewer 1 Report (Previous Reviewer 1)

As I mentioned in the previous revision, although it has some shortcomings, I believe that it is an article that will shed light on the literature. Therefore, I think it would be appropriate to accept it.

Author Response

Thank you.

Reviewer 2 Report (New Reviewer)

This study employed the immunohistochemistry method to identify the TRPC family's presence in the human cochlea, revealing different expressions and locations of the TRPC family through this technique. While the research and methodology in this study are straightforward, the author has provided comprehensive background information.However, there are several concerns regarding this manuscript. Firstly, the authors only mention the TRPC6 antibody being tested with control, leaving questions about the verification procedures for other antibodies. It would be helpful to explain the verification procedures for all antibodies used in this study.

Secondly, the sample preparation is unclear and does not sufficiently support the results. The localization is not clear through IHC methods, as the authors only use myosin as a reference to define the hair cell position. In my opinion, the results are not clear enough to illustrate the expression level in a specific region. it may be beneficial to use IF staining  as comparison methods in the study to provide more precise localization and better support for the results.

Author Response

Thank you for helpful comments!

Immunofluorescence was not performed but we hope our ms can be accepted for publication in the present form.

Reviewer 3 Report (New Reviewer)

This research is interesting and the article is well written and deserves to be published – The only observation is to change the order of the sections : in the first place “materials and methods” and then the “results and discussion”

Best regards

Author Response

Minor issues were done.

Reviewer 4 Report (New Reviewer)

This was a well designed and well presented study  that adds to the basic knowledge of how sounds are transmitted form the cochlea to ultimately the CNS.   In view of the recent studies concerning gene manipulation and the possibility of preventing or treating sensorineural deafness, papers like this are paramount to effort.

Author Response

Thank you very much!

Round 2

Reviewer 2 Report (New Reviewer)

This study used immunohistochemistry to identify the presence of the TRPC family in the human cochlea, revealing varied expressions and locations. The author addressed antibody verification concerns and no further issues were found. Although the research design can be improved, duplicating the experiment with human tissue samples is challenging. I agree to keep the current format. Additionally, the study provides comprehensive background information, which is beneficial for readers interested in this field.

This manuscript is a resubmission of an earlier submission. The following is a list of the peer review reports and author responses from that submission.

Round 1

Reviewer 1 Report

The authors investigated the presence of TRPC channels in human cochlea and they found immunopositivity for TRPC3, TRPC5 and TRPC6 in certain sections of the cochlear tissue. Transient receptor potential channels have diverse roles in mechanosensation and the physiological significance of these ion channels in sensory systems is rapidly emerging. Some studies revealed the presence and functionality of these channels in rodents but this is the first study conducted in human. It is very difficult and also important to reveal the presence of these channels in the human cochlea. Of course, the study has some weaknesses. For example, since the hearing levels of the donors are not known, it is not possible to comment on the physiological effects of these channels. Also, since the donors are all elderly people, there will likely be hearing problems. For these reasons, the functionality of these channels in people with healthy hearing remains a mystery. However, in the end, I believe that this study will shed light on future studies and contribute to the literature, as it is the first study to show the presence of these channels in the human cochlea.

Author Response

  • The authors investigated the presence of TRPC channels in human cochlea and they found immunopositivity for TRPC3, TRPC5 and TRPC6 in certain sections of the cochlear tissue. Transient receptor potential channels have diverse roles in mechanosensation and the physiological significance of these ion channels in sensory systems is rapidly emerging. Some studies revealed the presence and functionality of these channels in rodents but this is the first study conducted in human. It is very difficult and also important to reveal the presence of these channels in the human cochlea. Of course, the study has some weaknesses. For example, since the hearing levels of the donors are not known, it is not possible to comment on the physiological effects of these channels. Also, since the donors are all elderly people, there will likely be hearing problems. For these reasons, the functionality of these channels in people with healthy hearing remains a mystery. However, in the end, I believe that this study will shed light on future studies and contribute to the literature, as it is the first study to show the presence of these channels in the human cochlea.
  • Thank you very much for your comments. We appreciate your opinion on our work.

Reviewer 2 Report

Introduction: There are more than "a couple of gene variants" that are involved in SNHL". Since the authors are discussing TRPC channels, this has to do with calcium homeostasis, and the authors should make it clear that they will discuss TRPC variants, which are calcium channels. Also, there has been a lot of animal studies of TRPC in the cochlea, and their function in animals is quite relevant here.

line 53-64. Since this is an anatomical study, a picture and explanation of the anatomy would be helpful. Discussing a KCNQ1 channel is, I think, beside the point, since the gene of interest is a Ca++ ion channel. There are many other ion channels in the cochlea, particularly in the synaptic regions of OHC and IHCs, 

lines 74-83 should be in methods. 

2. Results and Discussion - 2.1 should go into Methods. 2.2 much of this should go in methods, and it is not necessary to tell the reader about the success or failure of decalcification or complications of a malfunctioning microtome. 

Figure 1 - doesn't look like immunostaining at all, looks like a routine H and E stain. This is poorly explained. What colors should these stains be?

line 157 - should be its own paragraph. Since the authors are discussing Hensen's, Deiters', and other cells, these should be pointed out in the pictures.

line 226 - mean age should be description of the specimens, perhaps in methods. 

Author Response

  • Introduction: There are more than "a couple of gene variants" that are involved in SNHL". Since the authors are discussing TRPC channels, this has to do with calcium homeostasis, and the authors should make it clear that they will discuss TRPC variants, which are calcium channels. Also, there has been a lot of animal studies of TRPC in the cochlea, and their function in animals is quite relevant here.
  • Thank you for your worthy comment. We adjusted and set a focus on TRPC and calcium.
  • Line 53-64. Since this is an anatomical study, a picture and explanation of the anatomy would be helpful. Discussing a KCNQ1 channel is, I think, beside the point, since the gene of interest is a Ca++ ion channel. There are many other ion channels in the cochlea, particularly in the synaptic regions of OHC and IHCs.
  • Thank you very much. You are absolutely right, for non-anatomists, a picture would truly facilitate comprehension. We therefore added two professional cochlea drawings. We discussed KCNQ1 in the context of endolymph production. But, again, you are right, many other molecular themes could be discussed and our excursion is therefore not contributing to the essence of the manuscript. We removed it.
  • Lines 74-83 should be in methods.
  • We can perfectly understand your request. However, the journal demands a short summary of the work with the principal conclusions in the introduction. This is why, we did not change these lines and hope that you, dear reviewer, will agree with us.
  • Results and Discussion - 2.1 should go into Methods. 2.2 much of this should go in methods, and it is not necessary to tell the reader about the success or failure of decalcification or complications of a malfunctioning microtome. 
  • Again, thank you for this suggestion, with which we completely agree. We therefore followed your instructions.
  • Figure 1 - doesn't look like immunostaining at all, looks like a routine H and E stain. This is poorly explained. What colors should these stains be?
  • Thank you very much for this key comment. We replaced some of the pictures by similar pictures with higher quality and added explanation to IHC coloration in the legends. Because the chromogen is red, it indeed looks a bit like standard HE staining.
  • Line 157 - should be its own paragraph. Since the authors are discussing Hensen's, Deiters', and other cells, these should be pointed out in the pictures.
  • First point: done and adjusted in the context of reformatting the manuscript structure to short communication type. Second point: Since these cell types cannot be identified with security in the figures, we removed these specifications in the text.
  • Line 226 - mean age should be description of the specimens, perhaps in methods. 
  • Agreed and done.

Round 2

Reviewer 2 Report

Introduction: lines 75-78 subject-verb disagreements. 

Results and Discussion - still confusing, needs better English. Figures are poorly fixed, for instance in Figure 3 - Organ of Corti is disrupted, there is no evidence of hair cells in this structure.